# Microbiome—Stealth Regulator of Breast Homeostasis and Cancer Metastasis

**DOI:** 10.3390/cancers16173040

**Published:** 2024-08-31

**Authors:** Saori Furuta

**Affiliations:** 1MetroHealth Medical Center, Case Western Reserve University School of Medicine, 2500 MetroHealth Drive, Cleveland, OH 44109, USA; sxf494@case.edu; 2Case Comprehensive Cancer Center, Case Western Reserve University School of Medicine, Cleveland, OH 44106, USA

**Keywords:** breast cancer, cancer metastasis, tumor-resident bacteria, microbiome, metabolites

## Abstract

**Simple Summary:**

It has long been known that breast tumors harbor various types of microbes. However, it was little known where these tumor-resident microbes came from and how they could contribute to breast cancer pathogenesis. Now, recent discoveries have unveiled that these tumor-resident microbes come from different parts of the body and live inside tumor cells to participate not only in tumorigenesis events, i.e., DNA damage and genomic instability, but also in tumor progression and metastasis. Such important findings have helped identify these intratumoral microbes as potential new targets for breast cancer treatment and prevention.

**Abstract:**

Cumulative evidence attests to the essential roles of commensal microbes in the physiology of hosts. Although the microbiome has been a major research subject since the time of Luis Pasteur and William Russell over 140 years ago, recent findings that certain intracellular bacteria contribute to the pathophysiology of healthy vs. diseased tissues have brought the field of the microbiome to a new era of investigation. Particularly, in the field of breast cancer research, breast-tumor-resident bacteria are now deemed to be essential players in tumor initiation and progression. This is a resurrection of Russel’s bacterial cause of cancer theory, which was in fact abandoned over 100 years ago. This review will introduce some of the recent findings that exemplify the roles of breast-tumor-resident microbes in breast carcinogenesis and metastasis and provide mechanistic explanations for these phenomena. Such information would be able to justify the utility of breast-tumor-resident microbes as biomarkers for disease progression and therapeutic targets.

## 1. Introduction

External and internal surfaces of animal bodies are entirely covered by microorganisms. For humans, each person contains about 40 trillion microbes, which is more than the total number of 30 trillion host cells [1]. These commensals are mostly found in the gut, where their density reaches almost 10 trillion microbes per milliliter, and they weigh about one kilogram per person [2]. It is increasingly evident that these symbionts are not merely passive passengers but essential players for fundamental functions of the body, including immunity, metabolism, and energy balance [3]. This notion is well supported by the fact that germ-free (GF) mice manifest serious defects in lymphoid tissue structure and functions [4]. In particular, their gut mucosal immunity is severely compromised due to low numbers of lymphocytes and antibody production [5,6,7]. 

It is increasingly evident that different physiological conditions between healthy cohorts and cancer patients could be largely attributed to discrete properties of microbial floras. It is established that not only tumor microbiota but also gut microbiota of cancer patients are far less diverse than the normal counterparts—a condition of ‘dysbiosis’. Thus, correcting the microbiome of cancer patients has gained traction as an adjuvant approach [8,9,10]. For example, normalizing the microbiome of cancer patients through FMTs from healthy cohorts or treatment-responders has been proven for its therapeutic benefits. A study by Di Modica et al. reported that transferring feces from Trastuzumab responders or non-responders to antibiotic-treated mice with HER2-positive breast cancer was able to reconstitute drug responsiveness [11]. However, the detailed mechanisms of FMT-induced anti-cancer effects have not been fully elucidated. 

Along with the gut microbiota, the breast microbiota is proposed to play critical roles in breast health and carcinogenesis [8,9,10]. Breast-resident microbes originate from either skin/nipple microbiota or are translocated from the gut along with immune cells, such as dendritic cells and macrophages [8,12,13]. Breast microbiota, however, could also be modified by environmental agents, such as aseptic solutions affecting skin microbiota [14]. Gut–breast microbial translocation, termed the ‘gut–breast axis’, greatly contributes to the composition of microbiota of breast tissues and milk [15,16,17,18]. This phenomenon, however, has mostly been conceived in relation to pregnancy and its influence by female sexual hormones [15,16,19,20,21,22]. Thus, it remains unclear whether the gut–breast axis exists outside pregnancy on a regular basis and contributes to breast pathophysiology. If this holds true, the gut–breast axis may contribute to FMT-mediated anti-breast cancer effects. 

## 2. Breast Tissue Microbiota

The human breast contains a unique microbiota different from those in other parts of the body, playing critical roles in breast health as well as the health of offspring [23]. The breast tissue microbiota is more diverse (higher α-diversity) than that of skin tissue, while species’ relative abundances (Shannon indices) are similar between them [24]. Sample-to-sample differences in microbiota compositions (β-diversities) are also higher in the breast compared to the skin, owing to major differences in less abundant microbial species [24]. These features of the breast microbiota are commonly found in individuals with different ages, nationalities, and parity statuses [23]. Based on the microbiota of healthy livers or breast tumors, healthy breast-resident bacteria are expected to mostly reside inside parenchymal cells [25,26]. However, there are clear differences between healthy breast microbiota and breast tumor microbiota [24]. For example, the most abundant bacterial phyla in healthy breast tissues of women older than 18 years old are *Proteobacteria* and *Firmicutes*, whereas these bacteria are under-represented in tumors (Table 1) [27]. Many of these bacteria which are abundant in healthy breast tissues produce beneficial biomaterials that confer anti-tumor and pro-immunogenic activities to protect the healthy tissue microenvironment.

## 3. Breast Milk Microbiota

Breast milk microbiota are proposed to be linked to breast tissue microbiota, although there has not been any study to confirm their direct relationship. Breast milk microbiota are detectable from the third trimester of pregnancy through lactation. Breast milk, in particular colostrum (the first milk after giving birth), is the primary source of commensals to the newborn [57], whereas maternal–neonatal microbial transfers during pregnancy are conducted through the placenta and amniotic fluid [58,59]. This bacterial transfer through breast milk greatly contributes to the bacterial composition of infants’ guts, which is similar to that of breast milk [60]. The breast milk microbiota plays critical roles in the infant’s immune development and his/her health in early and later life. Thus, dysbiosis of the breast milk microbiota would greatly influence infant development [61]. Typically, a baby ingests 1 × 10^5^ to 1 × 10^7^ bacteria a day while consuming approximately 800 mL/day of breast milk [62]. These breast milk bacteria include the genera *Lactobacillus*, *Staphylococcus*, *Enterococcus*, and *Bifidobacterium* (Table 1) [57]. Breast milk contains olisaccharides, which are indigestible by the host but are digested by enzymes produced by specific gut bacteria, such as *bifidobacterial* and *lactobacilli*, which utilize the metabolites for their expansion [63]. In addition, breast milk contains bacterial species (e.g., *Coprococcus*, *Faecalibacterium*, and *Roseburia* spp) that produce short-chain fatty acids (SCFAs), such as butylate, acetate, and formic acid. These SCFA-producing bacteria repopulate the neonatal gut and play a beneficial role in weight gain and adiposity [59,64,65,66]. 

## 4. Breast Tumor Microbiota

Breast cancer is the second leading cause of cancer-related deaths in women, with about 300,000 new cases per year in the US [67]. It is a heterogeneous malignancy, and distinct molecular subtypes have been characterized, including the Luminal A subgroup expressing the hormone receptors estrogen receptor (ER) and progesterone receptor (PR); the Luminal B type expressing ER and PR plus human epidermal growth factor receptor 2 (HER2); the HER2-positive type (HER2+/ER−/PR−); and triple-negative breast cancer (TNBC; ER−/PR−/HER2−). These molecular characterizations have been primarily utilized to determine treatment regimens for specific breast cancers [68]. However, because of the frequent resistance of breast cancer to such targeted therapies [69], it is becoming clearer that other therapeutic strategies need to be developed [70], and targeting breast tumor microbiota has recently gained traction [71].

Microbes within tumors are mostly localized within tumor parenchyma as well as immune cells [26,53]. Since normal-tissue microbes are also expected to reside within parenchymal cells, they are proposed to be a major source of intratumoral microbes [72,73]. Nevertheless, breast tumor microbiota are greatly different from healthy breast microbiota, indicating substantial influences of bacterial transfer from other parts of the body during tumorigenesis (Table 1). Breast tumor microbiota are in general abundant in *Fusobacterium*, *Atopobium*, *Gluconacterobacter*, *Hydrogenophaga*, and *Lactobacillus* [24], unlike normal breast microbiota abundant in *Proteobacteria* and *Firmicutes* [27]. The breast tumor microbiota is associated with dysregulation of cell proliferation, metabolic pathways, and immunological responses, contributing to tumor growth and progression (Table 1) [40,45,46,74,75]. Conversely, the normal breast microbiota is associated with increased cysteine and methionine metabolism, glycosyltransferases, and fatty acid biosynthesis, promoting immunological responses [24,76,77]. Furthermore, the breast tumor microbiota is enriched in *Enterobacteriaceae* and *Staphylococcus* compared to healthy breast microbiota [27]. Both bacteria are known to produce genotoxins that induce DNA damage to help induce malignant progression of host cells [38,39]. In addition, lactic acid-producing *Lactobacilli*, also abundant in breast tumor microbiota, could lower pH and induce metabolic rewiring of the tumor microenvironment (TME), leading to chemotherapy and radiation resistance of tumors [78]. These three taxa of breast-tumor-associated bacteria are also found to promote tumor metastasis and colonization, being transported along with tumor cells to the metastatic site [79]. In colorectal cancer, on the contrary, a different bacterial taxon, *Fusobacterium*, is transported along with colon cancer cells to the metastatic site [80], suggesting the roles of different bacteria in the metastasis of different types of cancers. The involvement of breast microbiota in tumor metastasis will be further discussed below.

### 4.1. Breast Cancer Subtype-Specific Microbiota

Different tumor types have distinct microbial compositions, indicating the impacts of different tissues/TMEs (Table 2) [26]. Furthermore, even among breast tumors, different tumor subtypes (Luminal A, Luminal B, HER2+, and triple-negative (TN) types) have distinct microbial compositions (Table 3). This indicates that the heterogeneity of molecular and metabolic profiles and cells of origin among different breast tumor subtypes impacts the fitness of different microbial communities [56,81,82]. For example, the phyla *Tenericutes*, *Proteobacteria*, and *Planctomycetes* are abundant in luminal subtypes (Luminal A and B). The most abundant genus in Luminal A tumors is *Xanthomonadales* (phylum *Proteobacteria*), while that for Luminal B tumors is Clostridium (phylum *Firmicutes*) [83]. Conversely, HER2+ breast tumors are abundant in *Akkermansia* (phylum *Verrucomicrobia*), *Thermi*, *Firmicutes* (*Filibacter*, *Anaerostipes*, and *Granulicatella*_*US31*), *Bacteroidetes* (*Cloacibacterium*, *Alloprevotella*, and *Dyadobacter*), and *Proteobacteria* (*Burkholderiales* and *Helicobacter pylori*, *PRD01a011B*, *Stakelama*, and *Blastomonas*) [9,26,83,84,85]. In contrast, TN breast tumors are enriched in *Euryarchaeota*, *Cyanobacteria*, *Firmicutes*, *Prevotella*, *Arcanobacterium*, and *Brevundimonas* [84,86]. In particular, the presence of *Listeria fleischmannii* (*Firmicutes*) in TN tumors is shown to be strongly associated with activation of the epithelial-to-mesenchymal transition (EMT) pathway, while the presence of *Haemophilus influenza* (*Pseudomonadota*) is correlated with tumor growth and cell cycle progression [87].

### 4.2. Race-/Ethnicity-Specific Breast Cancer Microbiota

Racial disparity is a clinical challenge of breast cancer. Breast cancer incidence is highest among White women; however, death rates are highest among Black women, likely due to distinct features of tumors allowing their aggressive growth and metastatic progression [111]. Above all, the immunological patterns of breast tumors are largely different between races (e.g., Asian: high levels of Th1 cells (IFNγ) and megakaryocytes; White: high levels of adipocytes, hematopoietic stem cells, and endothelial cells; and Black: high levels of activated dendritic cells, B cells, mesenchymal stem cells, and CXCL9 expression) [111]. 

The compositions of breast tumor microbiota also vary by race, which is proposed to contribute to racial differences in other tumor properties. Smith et al. showed that *Xanthomonadaceae* was the most abundant member in breast tumors from Non-Hispanic White women, whereas the genus *Ralstonia* was most abundant in breast tumors from Non-Hispanic Black women. They also showed that tumors from Non-Hispanic White women were richer in the Bacteroidetes phylum compared to Non-Hispanic Black women [83]. Similarly, Thyagarajan et al. reported that the Bacteroidetes phylum was over-represented in TN breast tumors from White women. Conversely, in TN breast tumors from Black women, the Actinobacteria and Thermi phyla and the *Bradyrhizobiaceae* genus were under-represented. TN tumors from Black women also showed a reduction in Shannon diversity compared to adjacent normal tissue, while the trend was reversed for White women [112]. Furthermore, Parida et al. reported racially distinct bacterial biomarkers for breast tumors. For Asian patients, *Pseudomonas*, *Terrabacter*, *Clostridiodes*, *Aestuariibacter*, *Succinimonas*, *Catellicoccus*, *Leucobacter*, *Rhizobium*, *Rhodococcus*, *Methylobacter*, and *Planctopirus* are elevated; for Black patients, *Xanthomonas*, *Amycolatopsis*, *Aphanizomenon*, *Anaerovorax*, *Aminiphilus*, *Trichormus*, *Chlorobium*, and *Sulfurovum* are elevated; and for White patients: *Halonatronum*, *Salinarchaeum*, and *Amorphus* are elevated. Such racially different bacteria produce distinct metabolites that could regulate different miRNAs and mRNAs of hosts, contributing to different levels of metastasis predictors (e.g., lung metastasis predictors, NMU, COL2A1, PRAME, and TTYH1, are highest in breast tumors of Black women) [111,113,114].

## 5. Origin of Breast Tissue Microbiota

The origins of breast tissues and milk microbiota are currently unclear; however, they are proposed to be derived from the breast skin, the oral cavity of the suckling infant, and the maternal gut through the gut–breast axis (Figure 1) [58]

### 5.1. Microbial Transfer from Breast Skin

Mechanisms of microbial transfer from the skin to the breast have not yet been clearly determined, although there are several possible scenarios based on the transfer of pathogenic bacteria from the skin to mammary glands. Abnormal microbiota of the breast skin could contribute to the pathogenicity of breast tissue, attesting to microbial transfer from the skin to the breast tissue [14]. Skin microbes such as *Pseudomonas aeruginosa* possess fatty acid-metabolizing capabilities and could become pathogenic in breast tissue [115]. Additionally, *Staphylococcus aureus* enriched in the skin of atopic dermatitis could lead to the formation of breast abscesses also colonized by *S. aureus* [116]. In fact, *Staphylococcus* is among the most abundant genera in breast tumors and strongly linked to breast cancer metastasis, attesting to the role of skin bacteria in breast tumors [40,47,112,117]. In particular, bacteremia, or colonization of S. aureus in certain tissues, could promote the incidence of primary tumors [118,119]. 

Other bacterial taxa linked to increased breast cancer risk include *Bacillus*, *Bacteroidetes*, *Brevundimonas*, *Comamonadaceae*, *Enterobacteriaceae*, and *Methylobacterium*, which are also found in the skin microbiota, supporting the possibility of their transfer from the skin [47,110,112,120]. Furthermore, increased numbers of *Corynebacterium* and *Pseudomonas*, usually only found in normal skin flora, could break the skin barrier and penetrate deep into the breast tissue to induce granular lobular mastitis [121,122]. 

Iatrogenic breakdown of the skin barrier during medical procedures could also result in contamination of the underlying breast tissue by skin commensals [123]. For example, breast skin microbes, such as *Staphylococcus epidermidis*, play roles in the pathogenesis of breast-implant complications, including anaplastic large-cell lymphoma [123,124,125,126,127]. On the contrary, mechanisms of microbial transfer from the skin to the breast tissue without damage to the skin barrier are more uncertain. Proposed scenarios include retrograde transfer through the nipple and ducts (see the details below) [57] and contamination during nipple aspirate fluid procedures [110]. 

### 5.2. Microbial Transfer from the Nipple

The nipple of the mammary glands contains about ten orifices of milk ducts [128]. It was initially proposed that these ductal openings facilitate bacterial transfer from the mother’s skin into the breast milk [129]. However, this possibility was ruled out by the finding that microbial compositions of nipple skin and nipple aspirate fluid are significantly different [130]. Specifically, there are strictly anaerobic species, such as *Lactobacilli* and *Bifidobacteria*, enriched in the breast milk which are unlikely to have originated from skin microbiota [29]. As a likely mechanism of bacterial transfer through the nipples, there are some degrees of retrograde flow of milk back into the mammary glands from the infant’s mouth during suckling [57,131]. Such oscillating milk flows allow mothers to respond to pathogens afflicting infants, build antibodies for them, and transfer these antibodies back to infants so that they can fight against illnesses. Nevertheless, such retrograde bacterial transfer from infants only serves to influence, not to act as the original source of, the maternal breast microbiota. In fact, certain anerobic bacteria, such as *Lactobacillus vini* and *L. paracasei*, are more abundant in the breasts of nulliparous or never-breastfed women than in those of breastfed women [35], suggesting that these bacteria are potentially derived from the maternal gut. 

### 5.3. Microbial Transfer via the Gut–Breast Axis or the Oro–Breast Axis

Translocation of gut bacteria to external tissues is commonly associated with disease conditions that impair the intestinal epithelial tight junctions allowing luminal bacteria to move across the epithelial barrier and get into the bloodstream [132]. However, such bacterial translocation also takes place, although to a lesser extent, in healthy individuals, involving beneficial gut bacteria such as *Lactobacilli* and *Bifidobacteria* [133,134,135]. Such non-disease-related bacterial translocation appears to involve select species and be associated with immunotraining and immunomodulation of the host [136,137,138]. 

During pregnancy and lactation, maternal gut bacteria translocate to the mammary glands so that they can be transferred to offsprings for colonizing their guts. During late pregnancy, there are synchronous changes in maternal mammary glands and guts. Mammary glands undergo structural and functional remodeling to become specialized organs that produce and transmit nutrients and other components necessary for neonatal growth [139]. Lactating mammary glands are also effector sites of the mucosal-associated lymphoid tissue system, playing essential roles in infants’ immunity [140]. Along with soluble immune factors, breast milk, especially colostrum, contains select types of leukocytes, such as neutrophils, macrophages, and lymphocytes [141], facilitated by the looser tight junctions of breast epithelia after giving a birth [142]. In particular, leukocytes exposed to antigens in the gut may migrate to the mammary glands and be transferred to infants through breast milk for their defense and immune training [143,144]. Synchronously, in maternal guts, epithelial permeability increases to facilitate bacterial transmigration [145]. Furthermore, the gut microbiota undergoes metabolic adaptation to elevated glucose levels, which would further modify their ability to translocate across the intestinal epithelium and reach the mammary glands [146].

Over decades, it has been known that breast milk, maternal feces, and infant feces share the same bacterial species, attesting to their physical connections [16]. Then, Perez et al. reported that a group of gut bacteria appeared to be physically translocated to mesenteric lymph nodes and then to the mammary glands during late pregnancy and lactation. Bacteria translocated to the mammary glands would then enter breast milk and be transferred to the infant to establish the microflora of the neonatal gut. Furthermore, the same study showed that viable gut bacteria found in milk-producing breast cells were also detected in peripheral blood mononuclear cells (PBMCs), indicating that PBMCs helped in the translocation of these gut bacteria to the breast [136]. 

The theory of gut–breast bacterial translocation was confirmed by studies demonstrating that orally administered *Lactobacilli* strains reached the breast milk of mothers [147,148]. Such studies also support another theory of oro–mammary bacterial translocation occasioned by the finding that maternal oral bacteria and milk microbiota partially overlap [149]. The majority of oral bacteria are expected to travel through the gastrointestinal (GI) tract to reach the gut and then be transported to the breast via the gut–breast axis. In contrast, a small fraction of oral bacteria could directly enter the oral and maxillofacial blood circulation to spread to distant tissues/organs [150]. Oro–mammary translocation is particularly important as a cause of the abundant oral bacteria (e.g., *Fusobacteria* and *Streptococci*) in breast tumors [150].

## 6. Mechanisms of Bacterial Translocation

Although the pathways and mechanisms by which certain bacteria enter the breast tissue have not yet been elucidated, some works have offered a plausible scientific basis. So far, there are two major mechanisms proposed: internalization/transcytosis by gut epithelia and direct sampling by phagocytic cells (Figure 2).

### 6.1. Internalization into Epithelial Cells

Luminal bacteria could be internalized in gut epithelial cells and subsequently taken up by dendritic cells (DCs) or macrophages in the mucosal environment. There are several potential mechanisms for the internalization of non-invasive bacteria into gut epithelial cells. First, intestinal epithelia harbor specialized microfold (M) cells that transcytose luminal bacteria to make them available to mucosal immune cells [151]. Alternatively, upon activation of TLR4, non-specialized enterocytes or kidney epithelial cells were found to transcytose Gram-negative gut bacteria [152]. Second, metabolic and oxidative stress, including hypoxia, low doses of nitric oxide, and uncoupling of mitochondrial oxidative phosphorylation, could damage the tight junctions of gut epithelia and induce transcytotic bacterial transfer to epithelia [153,154,155,156]. Third, low concentrations of IFNγ could cause the influx of noninvasive *E. coli* bacteria into human colon epithelia without affecting cell viability and tight junctions [157]. Such IFNγ-mediated transcytotic bacterial transfer was shown to depend on extracellular signal-regulated kinase (ERK) 1/2 and ADP-ribosylation factor (ARF)-6 [158]. Fourth, infection of intestines with the parasites *Giardia lamblia* or *Campylobacter jejuni* could damage gut epithelial barriers and tight junctions and induce penetration of luminal bacteria to the epithelia [159,160]. Fifth, viable non-pathogenic bacteria could enter host cells through endocytic pathways associated with lipid rafts and caveolin-1. Caveolin-1 or cholesterol was in fact found colocalized with bacteria-containing endosomes in epithelial cells [160,161]. These methods exploited by non-invasive bacteria are different from those of invasive pathogens that use specialized needle-like systems to inject effector proteins into epithelial cells and manipulate host cytoskeletons for anchorage and entry [162].

### 6.2. Sampling and Transportation by Immune Cells

It has been known that certain type of immune cells, especially those denoted as CD18+ cells, such as DCs and macrophages, could penetrate the gut epithelial barrier and directly take up non-pathogenic commensal bacteria from the lumen [163,164]. DCs especially are capable of opening tight junctions of intestinal epithelia and sampling luminal bacteria without destroying epithelial integrity because of their ability to repair damage. This allows non-invasive gut bacteria to spread to extraintestinal organs [163]. Similarly, macrophages could promote the extraintestinal dissemination of non-invasive gut bacteria [164]. It was shown that induction of DCs with viable commensal bacteria, but not dead bacteria, stimulates DC maturation, indicated by the increase in the class II major histocompatibility complex (MHC) and the B7.2 protein on the cell surface, and their translocation across the colon epithelium [165,166,167]. DCs, and possibly macrophages, that have taken up luminal bacteria would migrate to the nearby mesenteric lymph nodes and stay there for up to several days [168]. Such lymphoid-tissue-resident gut commensals are found to elevate anti-inflammatory signaling to help establish mutualism with host immunity [169]. Alternatively, these lymphoid-resident gut commensals could be taken up by lymphocytes and transported to distant tissues, such as lactating mammary glands [170,171]. In lactating mammary glands, the colonization of immune cells and their bacteria cargos is selective due to regulation by lactogenic hormones and retrograde signaling from suckling infants requesting specific immune cells to fight against their ailments [141,172,173].

## 7. Functions of Intracellular Microbiota

Increasing evidence demonstrates the existence of different intracellular bacteria in humans and mice [174,175,176,177]. The prevalence of intracellular bacteria over extracellular bacteria in tissues is largely attributed to more efficient immunological clearance of the latter than the former [178]. These intracellular microbes are proposed to play direct roles in the pathophysiology of normal tissues and tumors [179]. This view has been increasingly solidified since the groundbreaking discoveries of the roles of *Helicobacter pylori* in stomach ulcers and gastric cancer 30 years ago [180]. The recent surge in next-generation sequencing technologies has allowed investigators to profile microbial compositions of tumor tissues, identify tumor-associated microbes, and study their specific functions. Certain commensal bacteria invade cancer cells and remain inside cells during tumor progression and even metastasis [80,179]. This is largely attributed to the fact that intracellular bacteria are better protected than extracellular bacteria in a highly immunologic TME [79]. 

## 8. Bacterially Produced Metabolites

As discussed above, microbiota in human breast milk and breast tissue play essential roles in infants’ development and healthy intestinal microbiota and immunity. In particular, different bacterially produced metabolites contribute to differences in breast tissue microenvironments and the health of offspring [19]. A group of bacterially produced metabolites have been found to exert beneficial effects. For example, short-chain fatty acids (SCFAs), such as butyrate, acetate, and formic acid, in breast milk promote weight gain and adiposity in infants [66]. Cadaverine, a metabolite produced by bacterial lysine decarboxylase, is found to suppress breast cancer progression and metastasis, although the synthesis is downmodulated in breast cancer patients [181]. Also, indolepropionic acid (IPA) is a bacterial tryptophan metabolite that has cytostatic properties through activation of aryl hydrocarbon and pregnane X receptors. Ectopic application of IPA to breast cancer cells has been found to suppress their growth and metastasis [182]. On the other hand, another group of bacteria-derived metabolites exacerbate breast cancer growth. For example, queuine is a nucleobase mostly synthesized by certain pathogenic bacteria, such as *Clostridioides difficile* and *Chlamydia trachomatis*, to promote their virulence. Queuine is incorporated into specific transfer RNAs (tRNAs) which drive the expression of genes involved in cell proliferation and migration of breast cancer cells [183,184]. Furthermore, recent studies report anti-tumor effects of the bacterial metabolite trimethylamine N-oxide (TMAO), produced by a group of commensal bacteria, such as *Clostridia*, *Bifidobacteria*, and *Coriobacteria*. TMAO could promote the tumor-cell-killing activities of CD8+ T cells and M1-type macrophages. Analysis of clinical tumor samples found that TNBC tumors abundant in *Clostridiales* are enriched in TMAO and exhibit activated immune microenvironments [185]. 

## 9. Breast-Tumor-Associated Bacteria

The different levels of bacterial metabolites in normal vs. cancerous breast tissues discussed above are largely attributed to differences in microbial compositions. Decreased ratios of *Sphingomonas yanoikuyae* to *Methylobacterium radiotolerans* in the breast tissues are linked to elevated breast cancer risks [110]. *Lactobacillus*, *Staphylococcus*, and *Enterobacteriaceae* are more abundant in tumor-adjacent normal breast tissues compared to healthy breast tissues, indicating their contributions to neoplastic processes [47]. These different tumor-associated bacteria play differential roles in the development of breast cancer. Pro- and anti-tumor roles of select commensal bacteria are discussed below.

### 9.1. Origin of Breast-Tumor-Resident Bacteria

Breast-tumor-resident bacteria are proposed to have similar origins to those in normal breast tissues, namely, breast skin, the oral cavity of the suckling infant, the maternal gut through the gut–breast axis, and the maternal oral cavity through the oro–breast axis [186]. However, how these bacteria have traveled to distant tumors remains largely unknown. Bacterial strains found in tumors are mostly present in the gut microbiome, supporting the possibility of the gut–breast axis [187]. The oral microbiota is also one of the potential sources of breast-tumor-resident bacteria [188]. In particular, *Fusobacterium nucleatum*, a major human oral bacterium, is commonly found in breast tumor cells [30], while it is rarely found in the intestine and thus is expected to reach tumors through the circulatory system [189]. Furthermore, a study by Nejman et al. showed that bacteria in tumor-adjacent normal breast tissues had intermediate compositions between those of breast tumors and normal tissues [26]. This indicates that there are bacterial transfers between neighboring tissues that result in heterogeneity within tumors.

### 9.2. Major Breast-Tumor-Resident Bacterial Species

There are several major bacterial species frequently found in breast tumor samples.

#### 9.2.1. *Fusobacterium nucleatum*

*Fusobacterium nucleatum* is a common opportunistic bacterium in the oral cavity and is a potential causative agent of periodontitis and oral carcinomas [190]. This bacterium is also elevated in various types of solid tumors, especially in colorectal tumors, compared to matched healthy tissues [191]. It is also associated with liver metastasis, indicating the broad spread of this oral pathogen [80]. *F. nucleatum* localizes at tumor sites by attaching to cell-surface galactose-N-acetylgalactosamine (Gal-GalNAc) through its lectin Fap2 [29,179]. In particular, intravascularly injected Fap2-expressing *F. nucleatum* strain ATCC 23726 specifically colonizes mammary tumors in mice, whereas Fap2-deficient bacteria fail to do so. Furthermore, *F. nucleatum* secretes an amyloid-like filament FadA which not only helps the attachment and invasion of the bacterium to host cells [192], but also serves as the scaffold of biofilm formation and promotes cancer progression [193]. Within tumor cells, *F. nucleatum* induces pro-inflammatory signaling through the TNFα, NF-kB, and IL-6/IL-8 pathways [194,195,196] and promotes tumor growth, EMT, metastasis, and therapy resistance, while also suppressing NK cell-mediated tumor cell killing and T cell infiltration into tumors [29,98].

#### 9.2.2. *Streptococcus*

*Streptococcus* is an oral bacterium found to promote metastasis and colonization of metastasized breast cancer cells [79]. Streptococcus mutans is a Gram-positive bacterium associated with dental caries (cavities). This bacterium could invade endothelial cells through Toll-like receptor 2, which triggers the production of pro-inflammatory IL-6/IL-8 and monocyte chemoattractant protein-1 (MCP1). Inflamed endothelial cells elevate the permeability of blood vessels, leading to various systemic conditions. For example, intravenously injected *S. mutans* has been shown to induce lung vascular inflammation (e.g., thrombosis) and promote breast cancer metastasis to the lungs [197,198]. In addition, *S. cuniculiIn*, originally isolated from the respiratory tract of wild animals [199], has been shown to promote the metastatic potential of tumor cells by reorganizing actin cytoskeletons to resist shear stress during invasion [79]. In contrast, another strain of *Streptococci* confers beneficial effects on breast cancer treatment. *S. salivarius*, an abundant probiotic bacterium found in breast milk, has been shown to suppress breast cancer growth when applied ectopically. Similarly, *S. pneumoniae*, the bacterium responsible for pneumonia and lung cancer [200], produces endopeptidase O (PepO) virulence protein. Ectopic administration of PepO to a mouse model of triple-negative breast cancer (TNBC) has been shown to activate TLR2/4 in tumor-associated macrophages and suppresses breast tumor growth [201]. 

#### 9.2.3. *Staphylococcus* and *Enterobacteriaceae*

*Staphylococcus* and *Enterobacteriaceae* are intestinal bacteria that could induce DNA damage within host cells. *Staphylococci* produce a toxin, alpha phenol-soluble modulin (PSMα), and specific lipoproteins (Lpls). PSMα could induce DNA damage, whereas Lpls dampen DNA damage repair signaling, compromising the genomic integrity of the host cell [39]. Furthermore, *S. aureus*, a bacterium usually found in the upper respiratory tract and the skin, lowers the immunogenicity of the TME by suppressing effector T cells and promoting regulatory T cells [202]. In addition, *S. xylosus*, a skin commensal, promotes the metastatic potential of tumor cells by reorganizing actin cytoskeletons to resist fluid shear stress (FSS) during invasion [79]. Enterobacteriaceae (e.g., *E. coli* and *Salmonella*) are mostly intestinal commensals, and systemic infection with these bacteria is a common complication in cancer patients [203]. Similar to *Staphylococcus*, *Enterobacteriaceae*, especially those harboring the polyketide synthase (pks) island, produce a genotoxin, Colibactin, that causes DNA double-strand breaks [204]. *Enterobacteriaceae* also impair the expression of p53 tumor suppressor upon DNA damage, contributing to genomic instability of the host cell [205]. Furthermore, *Enterobacteriaceae*-infected host tumor cells produce bactericidal lysophosphatidylcholines, which have been found to be elevated in breast tumors and promote tumor growth and metastasis [206,207]. 

### 9.3. Roles of Intracellular Microbes in Breast Tumor Initiation/Development

According to the International Agency for Research on Cancer (IARC), 18–20% of cancers are caused by biological carcinogens such as oncogenic viruses and bacteria [208]. The roles of these microbes in cancer initiation and development involve six major mechanisms: genome instability/mutation, epigenetic modification, chronic inflammation, immune evasion, metabolic regulation, and metastasis [209]. Among these, we will specifically focus on the roles of microbes in genome instability/mutation and metastasis.

#### 9.3.1. Genome Instability/Mutation

The induction of genomic instability and mutation is one of the major carcinogenic mechanisms of microbes. Oncoviruses are some of the major breast-tumor-causing agents, including human papilloma virus (HPV), mouse mammary tumor virus (MMTV), Epstein–Barr virus (EBV), and bovine leukemia virus (BLV) [210]. They integrate the viral genome into the host chromosome to induce genetic mutations, while oncoproteins are produced by the integrated viral genome. For example, the HPV E7 oncoprotein directly inhibits the cGas-STING pathway involved in the expression of type I interferon and pro-inflammatory factors, leading to immune escape [211,212]. EBV LMP1 oncoprotein upregulates oncogenic signaling pathways, such as the NF-κB pathway, involved in cell proliferation [213], while MMTV oncovirus-infected cells escape apoptosis by activation of the Src tyrosine kinase pathway [214]. 

Likewise, certain carcinogenic bacteria, such as pks+ *Escherichia coli* and *Bacteroides fragilis*, secrete carcinogenic toxins that induce DNA damage, which results in elevated tumor onset and mortality [215]. The toxin produced by *Bacteroides fragilis* also promotes the expression of the enzyme spermine oxidase, producing reactive oxygen species (ROS) that cause DNA damage [216]. The oncobacterium *Fusobacterium nucleatum* secretes FadA, a key adhesin, that activates the E-cadherin/β-catenin pathway to upregulate checkpoint kinase 2 (CHK2), inducing DNA damage [217]. *Fusobacterium nucleatum* infection also downmodulates the Ku70/p53 DNA damage repair pathway, exacerbating DNA double-strand breaks (DSBs) [218]. *E. coli* and *Staphylococcus epidermis* isolated from breast tumors could cause DSBs even in cervical cancer cells, demonstrating non-tissue-specific tumorigenicity [47]. Furthermore, *H. pylori* and *E. coli* expressing EspF effector protein could suppress DNA mismatch repair mechanisms, augmenting genome instability and tumorigenesis [215,219]. 

Bacterial metabolites could also induce DNA damage to promote tumor development. Breast tumor tissues contain elevated levels of β-glucuronidase, a carcinogenic enzyme [130,220], that generates reactive intermediates from 2-amino-3-methylimidazo [4,5-f]quinoline to induce DNA damage [221]. Furthermore, *Streptococcus anginosus* and *Porphyromonas gingivalis* can convert ethanol to acetaldehyde, which could form DNA adducts or inhibit DNA repair enzymes, causing DNA damage [222,223].

#### 9.3.2. Tumor Metastasis

Over the past decade, it has been unveiled that intratumoral bacteria play critical roles in tumor metastasis. The initial study by Bullman et al. reported that primary colorectal tumors and their metastases shared the same viable bacterial components and that these bacteria were able to promote tumor cell growth and survival [80]. Furthermore, recent studies demonstrated that these tumor-resident bacteria are in fact localized in the cytosol of tumor cells and transported to metastatic sites by tumor cells. During tumor cell metastasis, the intracellular bacteria promote tumor cell survival by allowing them to overcome physical and biochemical hurdles in unfavorable environments through adaptations termed pro-metastatic processes [72,79,224]. During pro-metastatic adaptations, tumor cells acquire capabilities of breaking tissue boundaries, controlling the local environment, conferring immune suppression and resistance to mechanical stress, and remodulating tumor cell intrinsic properties, such as EMT, stemness, and adhesion (Figure 3) [224]. Here are some examples of intratumoral bacteria playing roles in the pro-metastasis and metastasis of tumor cells.

*Listeria* food bacteria, in particular *Listeria monocytogenes*, are intracellular pathogens found in decaying food [225]. *L. monocytogenes* has been long utilized to develop cancer vaccines because it induces potent innate and adaptive immunological responses [226]. However, this bacterium could also reside within breast tumor cells and promote the growth and metastasis of tumor cells, worsening the prognoses of patients [227,228]. Intratumoral *L. monocytogenes* induces cytoskeletal reorganization of tumor cells through its actin nucleation protein ActA and promotes tumor cell survival under FSS in the circulation [228]. Furthermore, such actin nucleation also recruits the ubiquitin-conjugating enzyme Ube2N that activates TAK1-p38 MAP kinase signaling that controls tumor cell metastasis [229,230]. Not only the pathogenic strain, but also a non-pathogenic strain, *L. fleischmannii*, resides within breast tumors and is strongly associated with elevated expression of EMT-associated genes [87].

Another tumor-metastasis-associated bacterium is *Fusobacterium nucleatum*, an opportunistic bacterium usually found in the oral cavity but which is also abundant in breast tumors [29]. Intratumoral *F*. *nucleatum* promotes tumor cell invasion and suppresses immunological response through several different mechanisms [29]. First, *F. nucleatum* produces a virulence factor, FadA, an amyloid protein that helps the binding of the pathogen to host cells [231]. FadA upregulates Mir4435-2HG, which then induces the expression of SNAIL1 triggering EMT of host cells [98]. Second, *F. nucleatum* elevates the expression of MMP-9, which degrades the extracellular matrix to assist tumor cell invasion. Third, *F. nucleatum* upregulates the expression of an adhesion molecule, ICAM1, through the ALPK1/NF-κB axis that promotes tumor cell adhesion to endothelial cells during intravasation [232]. Fourth, *F. nucleatum* induces the production of extracellular vesicles that promote the expression of TLR4 in neighboring tumor cells to help their growth and metastasis [233]. Fifth, *F. nucleatum* elevates the expression of immune checkpoint receptors, TIGIT and CEACAM1, that suppress immunological responses. Lastly, *F. nucleatum* directly invades and kills tumor-infiltrating lymphocytes, including NK cells and T cells [29].

The third tumor-metastasis-associated bacterium is *Bacteroides fragilis*, an abundant commensal bacterium in colon and breast tumors. *B. fragilis* produces an enterotoxin, *B. fragilis* toxin (BFT), a zinc-dependent metalloprotease commonly associated with inflammatory colon diseases. Intratumorally produced BFT could elevate the growth and metastatic potential of breast tumor cells by inducing the expression of stem cell-/EMT-associated genes such as Slug and Twist [111,234,235]. Regarding the mechanisms of this phenomenon, it has been shown that BFT induces the cleavage of E-cadherin on the surface of tumor cells, which then triggers nuclear localization of β-catenin and Notch effector NICD. Activation of both the Wnt and Notch signaling pathways greatly promotes stemness and metastasis in tumors [236].

Furthermore, *Staphylococcus* and *Lactibacillus*, commensal bacteria abundant in breast tumor cells, have been shown to translocate to the lungs along with metastasizing tumor cells. These intracellular bacteria inhibit RhoA/ROCK-induced contractility of tumor cells while being exposed to FSS, conferring protection against mechanical-force-induced apoptosis of tumor cells during metastasis [79]. As a potential mechanism, the same group proposed the possible involvement of ADP-ribosyltransferase C3 exoenzyme produced by these bacteria. This enzyme inhibits Rho GTPases to counteract immune cell activities and is well studied as a virulence factor of select bacterial species, such as *Staphylococcus* and *Bacilus* [237,238].

## 10. Discussion

Cumulative evidence unveils that intratumoral microbes are not only mere biomarkers for breast cancer phenotype and prognosis, but also the causes of breast cancer initiation and metastasis. Such roles of intratumoral microbes suggest that they could serve as potential targets for breast cancer treatment and prevention. Thus, the efficacy of antibiotics in combination with chemotherapy has been tested for breast cancer treatment. However, these antibiotics are reported to show both positive and negative effects, depending on whether they target tumor-resident bacteria or intestinal microbes [239]. For example, a Phase II study combining Moxifloxacin, a fourth-generation quinolone with broad-spectrum coverage of breast-tumor-resident bacteria, and a treatment of the physician’s choice (TPC; capecitabine, eribulin, gemcitabine, paclitaxel, or nab-paclitaxel) reported a promising efficacy and well-tolerated toxicities in patients with metastatic breast cancer [240]. On the other hand, a retrospective study on 772 women with triple-negative breast cancer treated with antimicrobials along with standard cytotoxic chemotherapy found that these patients had overall poorer survival than those without antimicrobials [241]. Such deleterious effects of antimicrobial use for breast cancer patients are largely due to intestinal microbiota disorders that impair immune function and trigger a systemic inflammatory response [239]. One possible solution to such a conundrum is to repopulate beneficial bacterial flora by supplementing probiotics and prebiotics to cancer patients treated with antibiotics. Furthermore, more recent strategies include fecal matter transplant from healthy cohorts to patients with breast cancer resistant to standard chemotherapy. Therapeutic manipulation of the tumor microbiome is an emerging research field which will revolutionize cancer therapy in the near future.

## 11. Conclusions

In summary, recent studies have unveiled that intratumoral microbes play critical roles in breast cancer development and metastasis, serving as potential biomarkers and therapeutic targets for the disease. Furthermore, given that these pro-tumor bacteria are likely derived from other parts of the body, including the skin and oral cavity, a future endeavor aiming to prevent their colonization and translocation to the breast tissue would warrant further investigation as a novel strategy for breast cancer prevention.

## Figures and Tables

**Figure 1 cancers-16-03040-f001:**
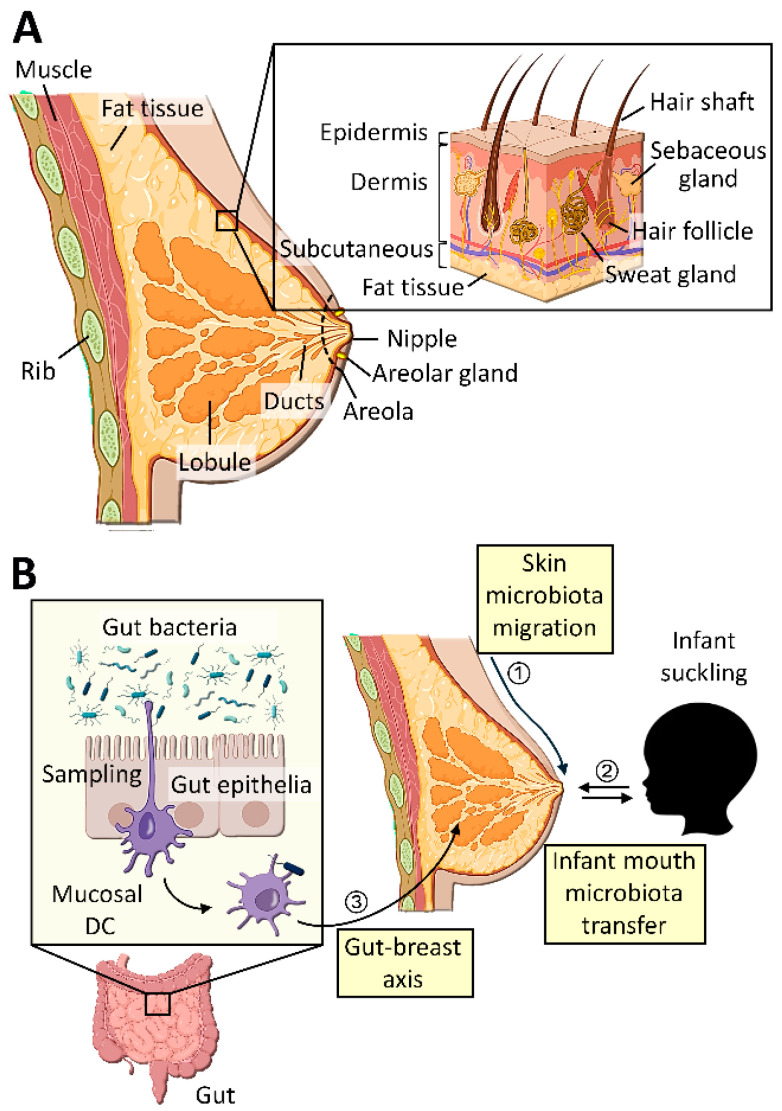
Potential origins of breast tissue/milk microbiota. (**A**) The structure of the human breast. Inset: the structure of the skin. (**B**) Three potential routes of microbial transfer to the breast milk/tissue. (1) Breast skin microbiota migration. Microbes of the adjacent skin could enter the mammary gland through the areola. (2) Infant mouth microbiota transfer. During suckling, the oral microbes of the infant could enter the mammary gland through retrograde transfer. (3) Gut–breast axis. Gut mucosal dendritic cells (DCs) occasionally sample commensal bacteria in the lumen and transfer them to lymphoid tissues and eventually reach the mammary gland.

**Figure 2 cancers-16-03040-f002:**
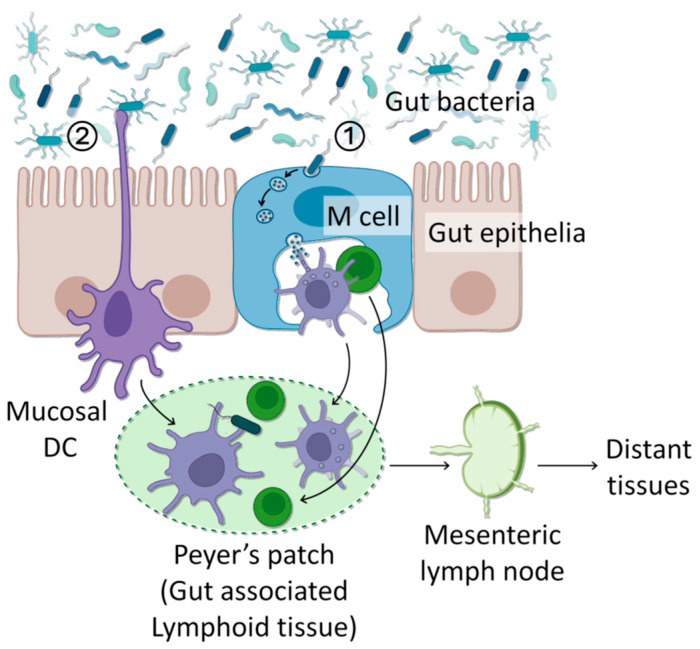
Two major modes of bacterial translocation. (1) Luminal bacteria are internalized by specialized microfold (M) cells present in the gut epithelial layer. These bacteria are transcytosed and made available to mucosal immune cells. (2) Luminal bacteria are directly taken up by CD18+ phagocytic cells in the mucosal layer that penetrate the gut epithelial layer. These immune cells which have taken up bacteria migrate to gut-associated lymphoid tissues (GALTs) and then mesenteric lymph nodes, where they stay until they are disseminated to thymus, spleen, and other distant tissues.

**Figure 3 cancers-16-03040-f003:**
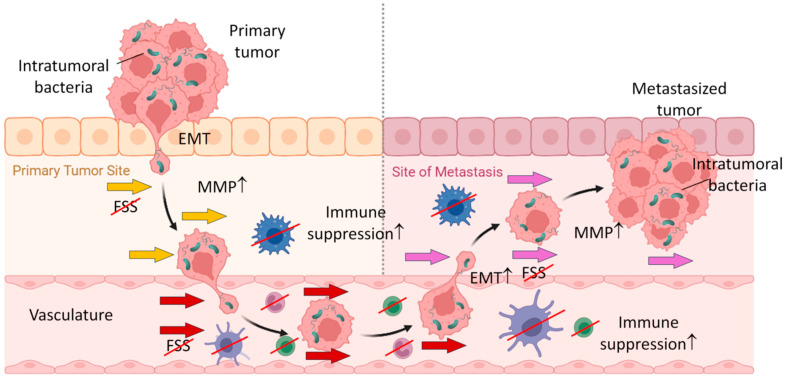
Roles of intratumoral bacteria in tumor metastasis. Bacteria inside the primary tumor could be transported to the metastasized tumor along with tumor cells. During metastasis, intratumoral bacteria assist the process in several different aspects. These include promotion of EMT, production of MMP, resistance to fluid sheer stress (FSS), and immune suppression. Arrows in different colors (yellow, red, and pink) indicate FSS at different locations.

**Table 1 cancers-16-03040-t001:** Bacteria differentially represented in normal vs. tumorous breast tissues.

Normal Breast				Breast Cancer			
Microbes	Levels	Functions	Ref.	Microbes	Levels	Functions	Ref.
*Sphingomonas*	Higher	Degrades environmental carcinogens, aromatic hydrocarbons, and polycyclic aromatic hydrocarbons; protective against ER+ breast cancer	[24,28]	*Fusobacterium nucleatum*	Higher	Promotes breast cancer cell attachment, invasion, and colonization during metastasis; impairs immunity and therapy response; activates β-catenin-mediated oncogene transcription and cell proliferation; produces β-lactamase for resistance to β-lactam antibiotics (e.g., penicillin)	[24,29,30,31]
*Firmicutes*, *Actinobacteria*	Higher	Negatively correlate with stromal fibrosis and breast cancer risk; enriched in breast milk	[32,33,34]				
*Lactobacillaceae*, *Acetobacterraceae*, *Leuconostocaceae Xanthomonadaceae*	Higher	Induce fructose and mannose metabolism and immune-related genes; enriched in breast milk of healthy women	[35,36,37]	*Enterobacteriaceae*, *Staphylococcus*	Higher	Induce DNA double-strand breaks in host cells	[38,39]
				*Ralstonia*	Higher	Dysregulates genes involved in carbohydrate metabolism	[35]
*Cyanobacteria*	Higher	Produce anti-cancer molecules (e.g., Cryptophycin F)	[40]	*Atopobium*, *Gluconacetobacter*	Higher	Modulate immunological responses	[24,41,42]
*Proteobacteria*, *Synergistetes*, *Tenericutes*	Higher	Regulate milk composition and production	[43,44]	*Porphyromonadaceae*, *Ruminococcaceae*	Higher	Participate in aberrant host metabolism	[40,45,46]
*Prevotellaceae,* *Butyricimonas*	Higher	Produce short-chain fatty acids (SCFAs) (propionate and butyrate) that exert anti-tumor activities	[40,47,48,49]	*Sutterella*, *Verrucomicrobiaceae*	Higher	Also found in cecal microbiota	[40,50,51]
				*Acinetobacter*	Higher	Abundant in HR+ and HER2+ breast cancer	[40,52]
				*Flavobacterium,* *Hydrogenophaga*	Higher	Abundant in metastatic breast cancer	[40,53,54]
*Alcaligenaceae*, *Moraxellaceae*, *Parabacteroides*	Higher	Enriched in breast milk	[40,55]	*Akkermansia* (*phylum Verrucomicrobia*), *Thermia*	Higher	Abundant in TNBC	[40,56]

**Table 2 cancers-16-03040-t002:** Representative bacteria in different types of tumors.

Cancer Types	Microbes	Levels	Pro-Tumor Mechanisms	Ref.
Breast	*Fusobacterium nucleatum*	Increased	Suppresses T cell infiltration into tumors; promotes tumor growth and metastatic progression	[29]
	*Anaerococcus*, *Caulobacter Propionibacterium*, *Streptococcus*, *Staphylococcus*	Decreased	Positively correlated with oncogenic immune features and T-cell activation-related genes	[9]
Bile duct	*Bifidobacteriaceae*, *Enterobacteriaceae*, *Enterococcaceae*	Increased	Increased production of bile acids and ammonia, leading to DNA damage in host cells and carcinogenesis	[88]
Cervical	*Fusobacterium* spp.	Increased	Associated with increased IL-4 and TGF-β1 mRNA in cervical cells	[89]
	*Anaerotruncus*, *Anaerostipes*, *Atopobium*, *Arthrospira*, *Bacteroides*, *Dialister*, *Peptoniphilus*, *Porphyromonas*, *Ruminococcus*, *Treponema*	Increased	Elevates vaginal pH to weaken host defense against infection and promotes tumor formation	[90]
Colorectal	*Bacteroides fragilis*	Increased	Increased interleukin-17 in the colon and DNA damage in the colonic epithelium, accelerating tumor onset and elevating host mortality	[91]
	*Fusobacterium*	Increased	Cancer cell proliferation and distant metastasis	[80]
Esophageal	*Lactobacillus fermentum*	Increased	Establishes acidic environment for growth advantage	[92]
	*Helicobacter pylori*	Increased	Spread from gastric colonization	[92]
	*Campylobacter* spp.	Increased	Causes inflammation that could contribute to carcinogenesis	[93]
	*Porphyromonas gingivalis*	Increased	Accelerates cell cycle and promotes cellular migration and metabolism of potentially carcinogenic substances such as ethanol to the carcinogenic derivative acetaldehyde	[94]
Extrahepatic Bile duct	*Helicobacter pylori*	Increased	Increases abundance of the virulence genes cagA and vacA and promotes tumor formation	[89]
	*Helicobacter bilis*	Increased	Induces inflammation to contribute to tumor formation	[95]
Gallbladder	*Fusobacterium nucleatum*, *Escherichia coli*, *Enterobacter* spp.	Increased	Promotes gallstone development and chronic cholecystitis to contribute to tumor formation	[96]
Gastric	*Helicobacter pylori*	Increased	CagA protein suppresses p53-mediated apoptosis of host cells while increasing cell motility and metastatic phenotypes	[97]
	*Fusobacterium nucleatum*	Increased	Induces epithelial-to-mesenchymal transition	[98]
Liver cancer	*Helicobacter bifidus*	Increased	Contributes to formation of chronic hepatitis that promotes tumor progression	[99]
Lung	*Acidovorax* spp.	Increased	Associated with carcinomas with p53 mutations	[100]
	*Thermus*, *Legionella*	Increased	Associated with advanced-stage and metastatic cancer	[101]
Oral cancer	*Fusobacterium nucleatum*	Increased	Induces epithelial-to-mesenchymal transition	[98]
	*Firmicutes* (*esp. Streptococcus*), *Actinobacteria* (*esp. Rothia*)	Increased	Elevated in normal oral tissues	[102]
Ovarian	*Mycoplasma*	Increased	Prevalent in 60% of tumors	[103]
Pancreatic	*Enterobacteriaceae*, *Pseudomonas* spp., *Mycobacterium avium*, *Pseudoxanthomonas*, *Streptomyces*, *Bacillus cereus*	Increased	Contributes to chemotherapy resistance and immune suppression	[104,105]
	*Malassezia globosa*	Increased	Induces the complement cascade through the activation of mannose-binding lectin C3 to promote tumorigenesis	[106]
Prostate	*Pseudomonas*, *Escherichia*, *Immunobacterium*, *Propionibacterium* spp.	Increased	Induces prostatitis and differentiation of prostate basal cells into ductal cells to promote tumor formation	[107]
	*Propionibacterium acnes* spp.	Increased	Induces prostatitis and promotes tumor formation	[108]
	*Staphylococcus*	Increased	Induces inflammation of the prostate tissue and promotes tumor formation	[107]
	*Fusobacterium nucleatum*, *Streptococcus oligosporus*	Increased	Induces chemoresistance by regulating autophagy	[109]

**Table 3 cancers-16-03040-t003:** Representative bacteria in different breast tumor subtypes.

Breast Cancer Subtypes	Microbes	Levels	Ref.
Luminal A	*Proteobacteria* (*Xanthomonadale*)	Increased	[83]
	*Tenericutes*, *Proteobacteria*, *Planctomycetes*	Increased	[110]
Luminal B	*Firmicutes* (*Clostridium*)	Increased	[83]
	*Tenericutes*, *Proteobacteria*, *Planctomycetes*	Increased	[110]
HER2+	*Thermi*, *Verrucomicrobia* (*Akkermasia*)	Increased	[83]
	*Firmicutes* (*Granulicatella:US31*), *Bacteroidetes* (*Dyadobacter*)	Increased	[26]
	*Firmicutes* (*Filibacter*, *Anaerostipes*), *Bacteroides* (*Cloacibacterium*, *Alloprevotella*), *Proteobacteria* (*PRD01a011B*, *Stakelama Blastomonas*)	Increased	[9]
	*Proteobacteria* (*Burkholderiales*, *Helicobacter pylori*)	Increased	[85]
TNBC	*Streptococcaceae*, *Ruminococcus*	Increased	[83]
	*Actinomycetaceae*, *Caulobacteriaceae*, *Sphingobacteriaceae*, *Enterobacteriaceae*, *Prevotellaceae*, *Brucellaceae*, *Bacillaceae*, *Peptostreptococcaceae*, *Flavobacteriaceae*	Increased	[86]
	*Prevotella*, *Brevundimonas*, *Actinomyces*, *Aerococcus*, *Arcobacter*, *Geobacillus*, *Orientia*, *Rothia*, *Streptococcaceae*, *Ruminococcus*, *Euryarchaeota*	Increased	[83,86]
	*Bartonella*, *Coxiella*, *Mobiluncus*, *Mycobacterium*, *Rickettsia*, *Sphingomonas*, *Azomonas*, *Alkanindiges*, *Proteus*, *Brevibacillus*, *Kocuria*, *Parasediminibacterium*	Increased	[68]

## Data Availability

All data generated or analyzed during this study are included in the published article.

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
