# Peer review of "Microbiome—Stealth Regulator of Breast Homeostasis and Cancer Metastasis"

_cancers, 2024, doi:10.3390/cancers16173040_

Round 1

Reviewer 1 Report

Comments and Suggestions for Authors

This review describes recent findings on breast resident microbes and what is known about their origin and contribution to breast physiology and pathology. Although research on this subject is still in its infancy, understanding this intricate interplay holds promises for developing innovative therapeutic approaches and the author has performed a significant and relevant effort to address this rapidly-growing research field. Although this article is well organized and several data were considered and discussed, few changes are suggested to make the manuscript easier to understand.

The paper concerning FMT stated at page 2 (lines 47-50) [Ref11] has not been reported correctly. This study, indeed, did not investigate FMT in patients, but feces obtained from responsive or non-responsive patients to trastuzumab treatment were used to re-constitute gut ecosystem of antibiotic treated mice.

Lines 69-70 - This sentence is not clear and should be re-written including more specifics.

Lines 171-172 - This sentence deserves the addition of more details elucidating how racial differences in breast tumor bacteria components may be linked to differential metastasis predictors among races.

Table 3 - Sample Type column should be deleted.

Some typos: e.g., “ant- tumor” instead of anti-tumor and “select bacteria” instead of selected bacteria.in line 363;” Heer” instead of Here in line 483 etc

FSS abbreviation should be specified in the text other than in Fig.3.

Author Response

General comment: The author appreciate the reviewer's extremely helpful comments to improve the paper quality.  The revised sections are highlighted in yellow.

1. The paper concerning FMT stated at page 2 (lines 47-50) [Ref11] has not been reported correctly. This study, indeed, did not investigate FMT in patients, but feces obtained from responsive or non-responsive patients to trastuzumab treatment were used to re-constitute gut ecosystem of antibiotic treated mice.

Response: This section has been re-written to describe the study finding correctly.

2. Lines 69-70 - This sentence is not clear and should be re-written including more specifics.

Response: This section has been rewritten for better clarity.

3. Lines 171-172 - This sentence deserves the addition of more details elucidating how racial differences in breast tumor bacteria components may be linked to differential metastasis predictors among races.

Response: This section has been rewritten to include more detailed information.

4. Table 3 - Sample Type column should be deleted.

Response: The table has been revised.

5. Some typos: e.g., “ant- tumor” instead of anti-tumor and “select bacteria” instead of selected bacteria.in line 363;” Heer” instead of Here in line 483 etc

Response: All the typos have been corrected.

6. FSS abbreviation should be specified in the text other than in Fig.3.

Response: FSS has now been spelled out in the text year Fig. 3.

Reviewer 2 Report

Comments and Suggestions for Authors

This paper primarily discusses the regulatory role of the microbiome in the occurrence and metastasis of breast cancer. While it has been previously acknowledged that various types of microbes are present in breast tumors, little is known about their origin within these tumors and their involvement in the mechanisms underlying breast cancer development. Recent studies have revealed that these microorganisms originate from different parts of the body and reside within tumor cells, actively participating in tumor initiation, progression, and metastasis processes. These significant findings contribute to identifying microbes within tumors as potential novel targets for both prevention and treatment strategies against breast cancer. Therefore, this article presents recent research findings that shed light on the role of microorganisms residing within breast tumors during breast cancer development and metastasis while providing a mechanistic explanation for these phenomena. This information effectively demonstrates how microbes residing within breast tumors can serve as biomarkers and therapeutic targets for disease progression.

1. The authors should provide an introductory section to elucidate the background of breast cancer, as well as elaborate on the current status of breast cancer and its primary treatment modalities.

2. The authors may refer to the following sources when providing the background information on breast cancer.

[1] He TJ, Zhao ZD, Luo ZT, Jia W, Zhang JT, Zhao Y, Xiao WC, Ming ZZ, Chen K. Advances in microbial decorations and its applications in drug delivery. Acta Materia Medica. 2023, 2(4): 466-479. DOI: 10.15212/AMM-2023-0036

[2] J. Li, A. Gu, X.-M. Nong, S. Zhai, Z.-Y. Yue, M.-Y. Li*, Y. Liu*, Six-Membered Aromatic Nitrogen Heterocyclic Anti-Tumor Agents: Synthesis and Applications. Chem. Rec. 2023, 23, e202300293.

3. The authors should enhance the summarization in the conclusion and provide further clarification on the trajectory of development.

4. The writer should meticulously review the grammar to ensure its accuracy.

5. The authors should ensure to provide an initial explanation of abbreviations upon their first appearance and consistently utilize them throughout the text.

6. The authors must meticulously verify the reference format to guarantee adherence to the journal's stipulations.

Comments on the Quality of English Language

Good

Author Response

The author appreciates the reviewer's constructive suggestions. The manuscript has been revised to incorporate all these points (revised sections are highlighted in blue).

Comment 1: he authors should provide an introductory section to elucidate the background of breast cancer, as well as elaborate on the current status of breast cancer and its primary treatment modalities.

Response 1:  I appreciate the reviewer's suggestion.  The manuscript has been revised to introduce the general information of breast cancer and  treatment modality.  This revised section is included under "4. Breast tumor microbiota" and highlighted in blue.

Comment 2: The authors may refer to the following sources when providing the background information on breast cancer.

[1] He TJ, Zhao ZD, Luo ZT, Jia W, Zhang JT, Zhao Y, Xiao WC, Ming ZZ, Chen K. Advances in microbial decorations and its applications in drug delivery. Acta Materia Medica. 2023, 2(4): 466-479. DOI: 10.15212/AMM-2023-0036

[2] J. Li, A. Gu, X.-M. Nong, S. Zhai, Z.-Y. Yue, M.-Y. Li*, Y. Liu*, Six-Membered Aromatic Nitrogen Heterocyclic Anti-Tumor Agents: Synthesis and Applications. Chem. Rec. 2023, 23, e202300293.

Response 2: I appreciate the reviewer's constructive suggestion. Both citations have been included in the section describing breast cancer landscape (highlighted in blue).

Comment 3: The authors should enhance the summarization in the conclusion and provide further clarification on the trajectory of development.

Response 3:  I appreciate the reviewer's suggestion.  The Conclusion section has been rewritten to enhance the trajectory of the development.

Comment 4: The writer should meticulously review the grammar to ensure its accuracy.

Response 4: The whole manuscript has been revised to correct grammatical errors.

Comment 5: The authors should ensure to provide an initial explanation of abbreviations upon their first appearance and consistently utilize them throughout the text.

Response 5: The whole manuscript has been revised to provide an initial explanation of abbreviations.

Comment 6: The authors must meticulously verify the reference format to guarantee adherence to the journal's stipulations.

Response 6: All the references have been revised to conform to the journal's formatting requirements.